

# Late Quaternary palaeoenvironmental evolution and sea level oscillation of the Santa Catarina Island (southern Brazil)

Lidia A. Kuhn[1,2], Karin A. F. Zonneveld[2], Paulo A. Souza[1], Rodrigo R. Cancelli[1]

[1]Laboratório de Palinologia Marleni Marques Toigo, Instituto de Geociências, Universidade Federal do Rio Grande do Sul, Porto Alegre, 91540-000, Brazil
[2]MARUM – Center for Marine Environmental Sciences and Faculty of Geosciences, University of Bremen, Bremen, 28359, Germany

*Correspondence to*: Lidia A. Kuhn (lidiaak.lak@gmail.com); Karin A. F. Zonneveld (kzonneveld@marum.de)

**Abstract.** Sea level oscillation during the Quaternary played a major role in the geomorphology and vegetation dynamics of coastal areas in southern Brazil, encompassing ecosystems that often have a unique biodiversity. Understanding the natural evolution of these areas is essential for decision-making of land use regulations towards sustainable development as well as to preserve the uniqueness of the coastal ecosystems. The southern Brazil coastal plain is formed by marine, transitional and continental Quaternary deposits controlled by relative past variations of the sea level. These variations shaped the coastal landscape and influenced the development of different Atlantic Rainforest formations, such as mangroves and restingas. In particular, the restinga formation corresponds to a specific ecosystem that covers sandy soils of marine and fluvial-marine origin formed during the Quaternary on the Brazilian coastal plain. In this contribution, we present high-resolution palynological and stable isotope data from a Holocene core retrieved from the coastal plain of the Santa Catarina Island (southern Brazil). We were able to identify four different environmental zones in the last 6500 yr BP. The first zone (6500–2820 cal yr BP) is characterized by a lagoon with large marine water influence. Notably, the observed dinoflagellate cyst association suggests that marine waters entering the region had its origin in the relatively warm saline Brazil Current waters. During the second zone (2820–1480 cal yr BP), marine water contribution to the lagoon decreased until it became disconnected with the sea. The third zone (1480–520 cal yr BP) was marked by the decrease of the water level until it dried out and led to the colonization of herbaceous vegetation over the palaeo-lagoon. The last zone (520 cal yr BP - recent) is characterized by the consolidation of the coastal plain Atlantic Rainforest (restinga vegetation). Our results form an example of the strong sensitivity of southern Brazilian ecosystem change caused by relative sea level variations. As such, it might contribute to the debate about potential effects of current climate change induced by global sea level variations.

## 1 Introduction

The comprehension of past environmental changes in the Quaternary is valuable for understanding modern and future environmental dynamics. This is particularly important in coastal areas where land-ocean interactions play a major role in the geomorphology and vegetation dynamics. This highly dynamic environment is often reshaped by anthropogenic activities





including the removal of vegetation, land use and hydrological changes. Such activities threaten the ecological and economical sustainability of the coastal areas that often have a unique biodiversity (Ramesh et al., 2015; Newton et al., 2016). The study of the natural evolution of these areas in terms of landscape and vegetation dynamics is essential to understand and preserve the uniqueness of these coastal ecosystems and support land-use regulations towards sustainable development of coastal areas.

Brazilian coastal areas and their ecosystems are under strong anthropogenic pressure. In such areas, the dominant biome is the Atlantic Rainforest, a global biodiversity hotspot recognized as one of the most important ecosystems of the earth (UNESCO, 2010). In particular, the restinga vegetation corresponds to a specific ecosystem within the Atlantic Rainforest that covers sandy soils of marine and fluvial-marine origin formed during the Quaternary on the Brazilian coastal plain (Scarano, 2002; Magnago et al., 2010). The southern Brazil coastal plain is formed by marine, transitional and continental Quaternary deposits,

controlled by the relative variations of sea level that directly influenced the development of different Atlantic Rainforest formations, such as restingas and mangroves (IBGE, 2012). A characteristic region for this system can be found in the south of Santa Catarina Island. This area hosts preserved fragments of the Atlantic Rainforest located near the coastline. It forms a particular region that allows the study of the interaction between sea level changes and vegetation dynamics.

The study of the past dynamics of coastal areas can be achieved by means of several scientific tools, such as sedimentological

(e.g., Dillenburg et al., 2006; Zazo et al., 2013), archeological (Martin et al., 1986), isotopic (e.g., Martin et al., 1986; Carr et al., 2015), paleontological (e.g., Angulo et al., 1999; Chemello and Silenzi, 2011; Toniolo et al., 2020), including palynological studies (e.g., Borromei and Quattrocchio, 2007; Leroy et al., 2013). Notably, the palynological records throughout sediment cores typically provide information on terrestrial and marine settings encompassing the environmental and vegetation changes within the same core (e.g., Mourelle et al., 2015; Kuhn et al., 2017). Additionally, the variations on relative abundances of

marine and continental palynomorphs are commonly used to determine sea level oscillations (e.g., van Soelen et al. 2010; Candel and Borromei, 2016).

Previous palynological studies were conducted in the southernmost portion of the southern Brazil coastal plain (i.e., Rio Grande do Sul coastal plain) (see summaries in Lorscheitter, 2003; Bauermann et al., 2009; Mourelle et al., 2018). However, the Santa Catarina coastal plain sector is geomorphologically distinct and similar studies are scarce, located only in the continental

portion (Behling and Negrelle, 2001; Amaral et al., 2012; Cancelli, 2012; Kuhn et al., 2017; França et al., 2019; Val-Péon et al., 2019; Cohen et al., 2020; Silva et al., 2021). In general, previous studies indicate a sequence of marine-influenced environments followed or not by lagoons and succeeded by terrestrialization. Nevertheless, most of the studies focused on the pollen record and the characterization of the marine-influenced and transitional environments were less explored.

This contribution provides the first high-resolution multi-proxy pollen, dinoflagellate cysts and isotopic study in the Santa

Catarina Island. We aim to reconstruct a detailed environmental evolution of the southern Santa Catarina Island (Fig. 1) throughout the Holocene to (i) understand the Atlantic Rainforest dynamics and consolidation in the Santa Catarina Island; (ii) identify the effects of Holocene sea level variations in the coastal landscape and vegetation evolution and (iii) compare with previous palynological studies regarding the environmental evolution of the southern Brazil Coastal Plain.







**Fig. 1**. Location and images of the study area. (a) Location of the Santa Catarina Island in southern Brazil (SC: Santa Catarina state; RS: Rio Grande do Sul state; Uy: Uruguay; Ar: Argentina). (b) Santa Catarina Island and location of the Pântano do Sul beach. (c) Shaded relief model of the Pântano do Sul beach and PCSC-4 core location (Basemap imagery source figures 1b and 1c: ArcGIS/ESRI/MAXAR; Shaded relief basemap source figure c: TOPODATA/INPE). (d) Panoramic aerial photo of the sampling location: note the preserved Atlantic Rainforest in the sampling area and the urban development.

**2 Environmental setting**

The Santa Catarina Island is located in the Santa Catarina sector of the southern Brazil coastal plain. The physiographic and structural aspects of the island are similar to the continental region since they were united when the sea level was below the current level (Horn Filho, 2006). The topography is dominated by granitic coastal mountains with altitudes of up to 532 meters and the coastal plain, which consists of Pleistocene and Holocene marine, beach, aeolian, lagoonal and paludal environment

deposits (Horn Filho, 2006). The formation of the latter deposit is associated with transgressive and regressive events of the relative sea level that occurred during the Quaternary (Caruso Jr, 1993).

The Santa Catarina Island is situated in a subtropical zone and the climate is characterized as humid oceanic without a dry season and with hot summers (Cfa, according to Koppen's classification) (Alvares et al., 2013). The Cfa climate type comprises maximum average temperature of more than 22ºC, minimum average between -3 and 18ºC, and rainfall is well distributed

along the year with annual accumulated precipitation of 1766 mm in the Santa Catarina Island (Alvares et al., 2013; INMET, 2022). The region is influenced by the South Atlantic Tropical Anticyclone and Polar Migratory Anticyclone. The South Atlantic Anticyclone produces the Atlantic Tropical Air Mass, a warm and humid mass that is active throughout the year, while the Polar Migratory Anticyclone generates the Atlantic Polar Air Mass, which is characterized by low temperatures and high humidity. The migration of the Polar Migratory Anticyclone to the region generates the polar front that is characterized

by unstable weather and increase of the precipitation (Nimer, 1990).

The Atlantic Rainforest covers a large portion of southern Brazil and the entire Santa Catarina coastal plain. This biome encompasses different forest formations and associated ecosystems. In the highlands (Serra Geral plateau), it is characterized as a mosaic of Araucaria Forest and grasslands, while in the coastal plain there are a dense arboreal vegetation and some pioneer formations such as the restinga, mangroves and salt marshes. These pioneer vegetations are conditioned by edaphic

factors and are composed by plants adapted to the ecological parameters of first occupation character (Oliveira Filho and Fontes, 2000; Scarano, 2002; Magnago et al., 2010; IBGE, 2012).

The Santa Catarina Island is bordered by the South Atlantic Ocean. Surface currents in the region are dominated by the southwards flowing Brazil Current (BC) (Fig. 2), relatively warm and saline (Peterson and Stramma, 1991). The BC originates at about 10° S from the bifurcation of the westward flowing South Equatorial Current (Silveira et al., 2000; Souza and Robison,

2004). The BC contacts the northward-flowing Malvinas Current (MC), which is characterized by cold and low salinity waters that have their origin in the Antarctic Circumpolar Current. At the contact zone, the so-called Brazil-Malvinas Confluence (BMC), MC waters dive under the BC waters. The mixed water masses are transported successively eastward as part of the

South Atlantic Current (Piola and Matano, 2019). Along the coast an additional northward flowing water mass can be observed: the Brazilian Coastal Current (BCC). This last current consists of low salinity waters discharged from the Rio de La Plata and

Patos Lagoon that, on its way north, mixes with the other water masses (Souza and Robinson, 2004). The positions of the maximal northward extension of the BCC as well as BMC vary strongly between the seasons (Piola et al., 2000, 2005). During austral summer, the BMC reaches its southernmost position. The maximal northern extension of the BCC as well as colder MC waters can be observed near the Santa Catarina Island in austral winter (e.g., https://podaac-tools.jpl.nasa.gov/soto/, "state of the ocean, temperature").

Below surface waters to a depth of approximately 600 m, southward flowing South Atlantic Central Water can be observed, which overlies the cold and less saline Antarctic Intermediate Water (Piola and Matano, 2001).

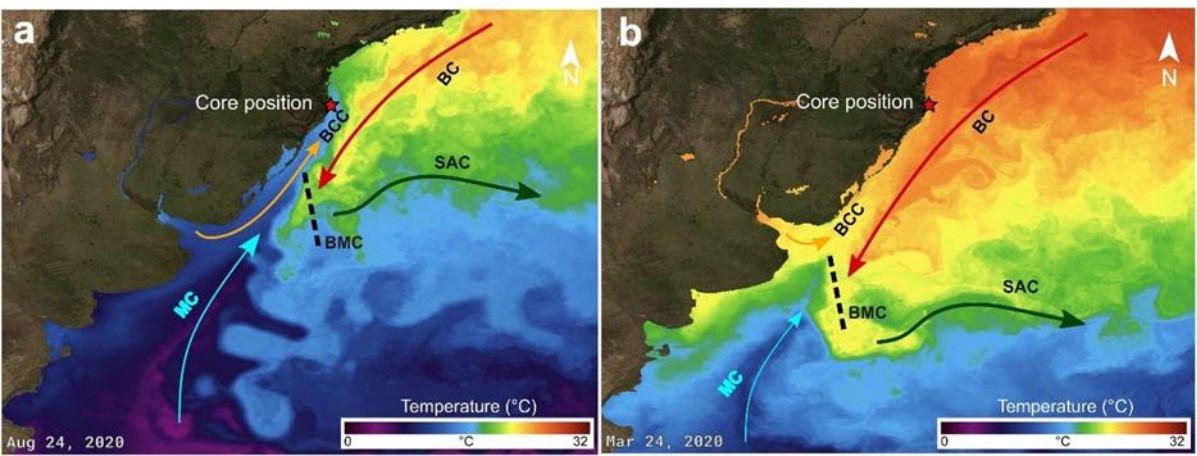

**Fig. 2.** Present configuration of sea surface currents and temperature in the western South Atlantic Ocean in the (a) winter and (b) summer (JPL MUR MEaSUREs Project, 2015). MC: Malvinas Current; BMC: Brazil-Malvinas Confluence; BC: Brazil

Current; BCC: Brazilian Coastal Current; SAC: South Atlantic Current.

## 3 Material and methods

### 3.1 Sediment core collection

This study was carried out on a sedimentary core (PCSC-4) retrieved from Pântano do Sul beach (27°46'36.49''S; 48°31'45.96''W; Fig. 1) located at the southernmost Santa Catarina Island. The core was drilled using a Russian Peat Corer

on a peat deposit, reaching a maximum depth of 650 cm. The sampling site is located approximately 1 km from the current coastline and ca. 1 m above the present sea-level. The 50 cm long sediments sections were sealed and transported to the Palynological Laboratory Marleni Marques Toigo at the Federal University of Rio Grande do Sul.





## 3.2 Radiocarbon dating

Four bulk organic rich sediment samples were selected along the core and analyzed with the Accelerator Mass Spectrometry

(AMS) at the CAIS Laboratory of the University of Georgia (USA) for radiocarbon dating. Sample selection was made after the palynological analyses aiming at obtaining ages for significant changes in the palynological record. Results were calibrated using the program CALIB Radiocarbon Calibration Version 8.2 (Stuiver et al., 2021), considering the Southern Hemisphere SHCal 13 radiocarbon calibration curve (Hogg et al., 2013). Interpolated ages were calculated using linear interpolation on Tilia software version 1.7.16 (Grimm, 2011) (Fig. 3).

## 3.3 Granulometric analyses


Grain-size analyses and calculation of organic matter content were made for 64 samples with 10 cm intervals along the core. The samples were equally separated into two sub-samples to determine the grain-size analyses and the organic matter content in the sediment. These analyses, as well as the calculations of the statistical parameters were carried out at the Center for Studies of Oceanic and Coastal Geology (CECO) at the Federal University of Rio Grande do Sul. All the samples were dried

in an oven at 40°C and then weighted. The analyses were performed by sieving-pipetting method, following statistical parameters of Folk and Ward (1957) and textural classification of Shepard (1954). To determine the organic matter content, the samples were calcined in muffle at 550°C during 4 h and weighed before and after the calcination. The organic matter content of the sediments was determined by the loss on ignition after this process.

## 3.4 Elemental C and N and $^{13}$C isotopes

Thirty-three samples with 3 cm³ of sediment were collected with 20 cm intervals from the top to the bottom of the core (0 - 640 cm) added to the basal sample (650 cm) for total organic Carbon (TOC), total Nitrogen (TN) and $\delta^{13}$C analyses. At first, the samples were dried in an oven at 60°C and weighted. Samples were treated with 10 % HCl to eliminate carbonate and then washed with Milli-Q water until the pH reached 5. Samples were dried in a freeze-dryer, weighted again, and homogenized to be analyzed in the Elemental Analyzer Coupled to Isotope Ratio Mass Spectrometry (EA-IRMS) in the Hinrichs Laboratory

at the University of Bremen. TOC and TN values are expressed as a percentage of dry weight and $\delta^{13}$C is expressed in delta per mil notation with an accuracy of ± 0.17‰, with respect to the VPDB standard. The C/N (weight/weight) was calculated using the elemental results ratio.

## 3.5 Palynological analysis

Sixty-six samples of 3 cm³ were obtained throughout the core for pollen/spore analyses with 10 cm spacing between them.

After a preliminary taxonomic recognition, samples where dinoflagellate cysts were recorded (650-310 cm) were resampled for more detailed analyses, totaling 35 samples with the same spacing and bulk volume of sample (3 cm³). One *Lycopodium*



*clavatum* L. spore tablet (18.584 ± 371 spores) was added to each sample before the chemical processing of both pollen/spore
and dinoflagellate samples to allow concentration calculations (Stockmarr, 1971).

The pollen/spore samples were processed following standard preparation techniques (Faegri et al., 1989), using HF (40 %),
HCL (10 %), KOH (10 %) and acetolysis. To concentrate the material, samples were sieved using a <250 μm sieve; $ZnCl_2$ was
used for heavy liquid separation checking the residues to be sure that no material was lost in the separation. Slides were
prepared from drops of the final residue, mounted with Entellan.

The dinoflagellate cysts samples were prepared using similar procedures. However, to avoid damage to the cysts, samples
were not prepared using hot acids, KOH and acetolysis. The dinoflagellate cysts samples were decalcified with diluted HCl
(10 %), treated with HF (40 %) to remove silicates. After chemical treatments, samples were sieved over a 20 μm mesh screen
and residues were transferred to an eppendorf vile where the material was concentrated in 1 ml. Slides were mounted with
glycerin jelly for microscopic analysis.

Pollen/spore samples were counted until reaching a minimum of 300 pollen grains monitored by saturation curves. The other
palynomorphs (i.e., spores, algae, acritarchs and microforaminiferal linings) and *L. clavatum* spores were counted in parallel.
Concentrations (palynomorphs/$cm^3$) were calculated using the *L. clavatum* spores as reference values.

Dinoflagellate cysts samples were counted until reaching their saturation curves. The total dinoflagellate sum adds all counted
dinoflagellate cysts and the relative abundances of each taxa in the dinoflagellate analyses are indicated as a percentage of the
total dinoflagellate sum. Concentrations (dinoflagellate cysts/$cm^3$) were calculated using the *L. clavatum* spores as reference
values.

To integrate the dinoflagellate cysts and pollen/spore counts, we used the ratio of *L. clavatum* counts from pollen/spore and
the corresponding dinoflagellate cyst samples as a conversion factor. The dinoflagellate cyst counts were multiplied by this
ratio and added to the final integrated diagram.

The total sum represents the sum of all palynomorphs (including dinoflagellate cysts), whereas pollen sum refers to the total
amount of pollen grains. The relative abundances of pollen grains were calculated as a percentage of the pollen sum whereas
relative abundances of the other palynomorphs were calculated in relation to the total sum.

The environmental zones were established from changes in the palynomorphs assemblages and from cluster analysis based on
percentage values of the total sum. The depth constrained cluster analysis (CONISS) was performed using the Edwards &
Cavalli-Sforza's chord distance square-root transformation. Cluster analyses, percentage and concentration diagrams were
constructed using the Tilia versions 1.7.16 (Grimm, 2011). For the Principal Component Analyses (PCA) we used the software
Canoco (Šmilauer and Lepš, 2014) and PAST 4.03 (Hammer et al., 2001). Multivariate analyses were performed on
palynological relative abundance data.

The taxonomic determinations of the pollen and spores were based on comparison with modern equivalents in palynological
reference collections ("MP-Pr" slides of the LPMMT/IGeo/UFRGS) and from the literature (e.g., Hooghiemstra, 1984, Neves
and Lorscheitter, 1992, Herrera and Urrego, 1996, Lorscheitter et al., 1998, Colinvaux et al., 1999, Macedo et al., 2009,
Cancelli et al., 2012). Dinoflagellate cysts were identified following the online key for dinoflagellate cyst determinations





(Zonneveld and Pospelova, 2015 and references therein). Dinoflagellate cysts were grouped according to their life strategies; photosynthetic taxa (*Operculodinium centrocarpum*, O. *israelianum*, *Spiniferites* spp., *Spiniferites mirabilis* and *Pentapharsodinium dalei*) and heterotrophic taxa (*Brigantedinium* spp., *Leipokatium invisitatum*, *Polykrikos kofoidii/schwartzii*, *Protoperidinium* spp. and *Selenopemphix nephroides*).

## 4 Results

### 4.1 Radiocarbon dating

The radiocarbon dating results are presented in Table 1, including uncalibrated and calibrated ages obtained from four selected samples. Calibrated ages indicate that the deposition of the studied core occurred entirely during the middle to late Holocene interval, once the lowermost level (650 cm) has an age of 6500 cal yr BP, whereas the uppermost level (55 cm depth) revealed an age of 390 cal yr BP. Remaining samples presented intermediate ages.

**Table 1.** Radiocarbon dates and calibrated ages of selected samples from the PCSC-4 core, south of Santa Catarina Island, southern Brazil. *Serial number of CAIS Laboratory of University of Georgia.

| Sample number* | Depth (cm) | Uncalibrated Age ($^{14}$C yr BP) | Calibrated age (cal yr BP) | Calibrated $^{14}$C age ($2\sigma$) (cal yr BP) |
|---|---|---|---|---|
| UGAMS# 49856 | 55 | $330 \pm 25$ | 388 | 357-446 |
| UGAMS# 49855 | 235 | $1750 \pm 20$ | 1614 | 1561-1631 |
| UGAMS# 49854 | 455 | $4730 \pm 20$ | 5394 | 5320-5427 |
| UGAMS# 35404 | 650 | $5760 \pm 20$ | 6503 | 6433-6568 |

### 4.2 Granulometric analysis

The core consists of unconsolidated sediments composed of medium sand, fine sand, silt, and clay added to variable amount of organic matter (Fig. 3). The organic matter is dominant (>80 %) from 220 cm to the top of the core. Regarding the distribution of clastic sediments, in general, there is a mixture of silt, clay and fine-to medium sand from the base up to 220 cm (see distribution in Figure 3). Localized calcareous shells in living position and shell fragments occur scattered from the base until 400 cm of depth.

### 4.3 Elemental C and N and $^{13}$C isotopes

The geochemical data are presented as individual profiles along the studied core (Fig. 3) and as the binary plot of $\delta^{13}C \times C/N$ (Fig. 4). Total organic carbon (TOC) concentration varies from 0.5 % to 49.6 % and shows two main intervals separated by a gradual transition between them. The interval from the base of the core up to 240 cm shows an average value of 5,8 % whereas between 220-200 the average is 24,5 %. Samples from 180 cm to the top present an average TOC value of 44,6 %. The total



nitrogen (TN) ranges from a minimum of 0.04 % at 650 cm depth to 2.1 % at 60 cm depth. C/N ratios (weight/weight) show

nearly constant values of ca. 15 from the base up to 240 cm, followed by an abrupt increase from 240 to 180 cm (15.2–47.8)

and a subsequent subtle decrease from 180 cm to the top (47.8–26.2) (Fig. 3). The $\delta^{13}$C results are in the range of -12.2 ‰ to

-29.6 ‰. The $\delta^{13}$C values are higher at the base of the sediment core (650–240 cm depth), with a range of -12.2 ‰ to -20.9

‰, followed by a downward trend towards the top of the core.

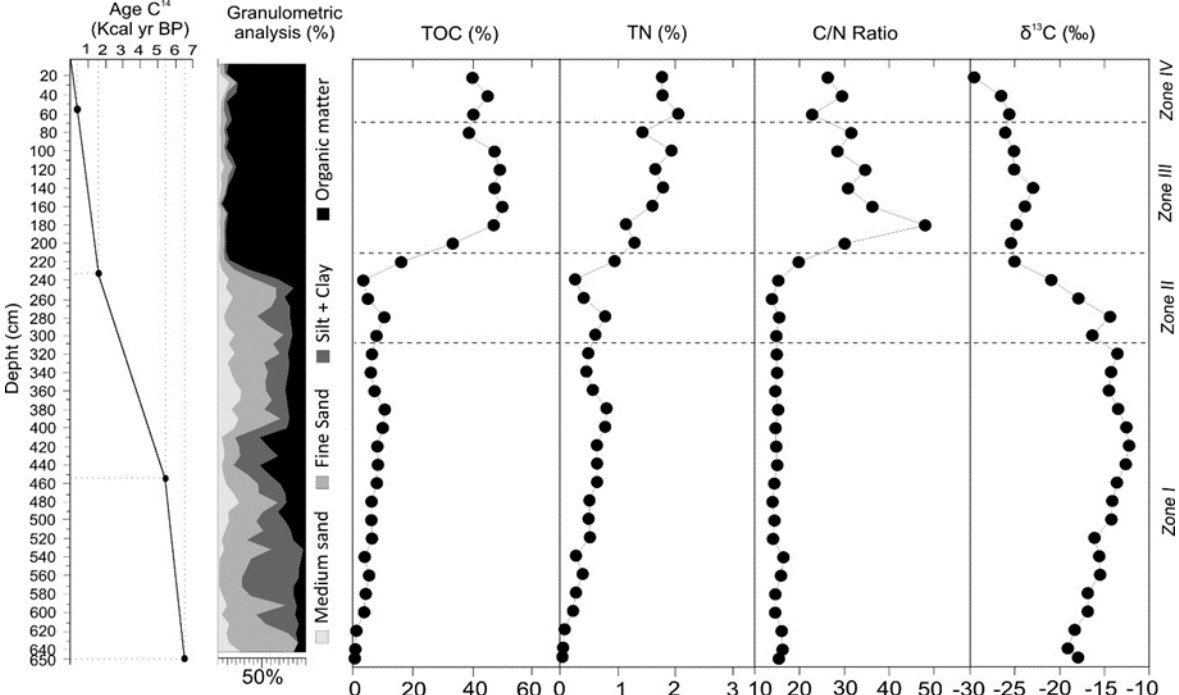

**Fig. 3.** Summarized results of the radiocarbon data, grain-size (granulometric) analyses, total organic carbon (TOC), total

nitrogen (TN), C/N ratio and $\delta^{13}$C values obtained from the PCSC-4 core, south of Santa Catarina Island, southern Brazil.





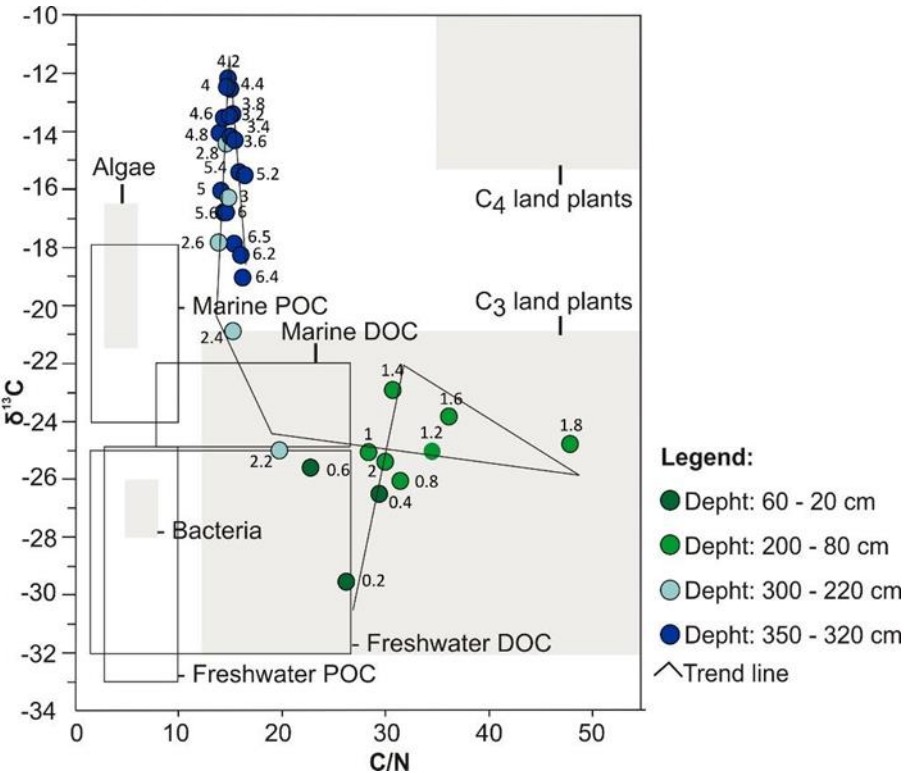

**Fig. 4.** Binary diagram illustrating the relationship between δ¹³C and C/N (according to Meyers, 1994 and Wilson et al., 2005) and the results obtained from each interval of depth of the PCSC-4 core, south of Santa Catarina Island, southern Brazil. DOC:

Dissolved organic carbon; POC: Particulate organic carbon.

### 4.4 Palynological record

A total of 114 distinct palynomorphs were identified along the 650 cm of the studied core, including pollen grains (59 taxa), spores (16), freshwater algae (4), marine algae (1), acritarchs (3), dinoflagellate cysts (10), indeterminate spores (8) and indeterminate pollen grains (13), as well as, fungi, microforaminiferal linings, escolecodons and copepod eggs.

The palynological diagrams show the distribution of palynomorphs in the samples, grouped according to their ecological affinities (habit or habitat) (Figs. 5-8). Both visual examination as well as the cluster and PCA analysis (Fig. 9) show the samples can be grouped in four zones with characteristic species associations. For simplification purposes, the results will be presented according to the four zones discussed in Section 5, in ascending stratigraphic order.

#### 4.4.1 Zone I (650-310 cm, sample 1-35)

This zone was recognized in the basal part of the core. It is characterized by high contents of marine palynomorphs (12-79 %), represented mainly by dinoflagellate cysts (8–78 %). Additionally, the other marine palynomorphs, i.e. microforaminiferal linings (<8 %), acritarchs (<17 %) and the prasinophyte *Cymatiosphaera* (<5 %), have the highest concentrations at this zone.





The dinoflagellate cysts include both photosynthetic (*Operculodinium centrocarpum*, *O. israelianum*, *Spiniferites* spp., *S. mirabilis*, *P. dalei*) and heterotrophic taxa (*Brigantedinium* spp. *Leipokatium invisitatum*, *Polykrikos kofoidii/schwarzii*,

*Protoperidinium* spp., *Selopemphix nephoides,* Fig. 8).

The acritarch association is composed of species of the genus *Micrhystridium*, Acritarch sp. 1 and Acritarch sp. 2. Freshwater algae are observed in low percentages through this zone and are mainly formed by *Botryococcus* spp., followed by *Spirogyra*, *Pseudoschizaea rubina* and *Zygnema*.

Terrestrial palynomorphs are dominated by pollen of trees and shrubs (34-63 %), followed by herbs (28-60 %), ferns (4–21

%) and indeterminate pollen grains (2-11 %). Arecaceae (4-26 %), *Alchornea* (4-20 %) and Myrtaceae (2-16 %) dominated the trees and shrubs pollen, whereas Poaceae (15 - 55%), *Amaranthus*/Chenopodioideae (1-17 %), Asteraceae (1-9 %) and Apiaceae (1-7 %) dominated the herbs pollen. Ferns are most represented by Polypodiaceae (3-13 %) and *Blechnum* (1-5 %). Epiphytes, lianas and climbers, bryophytes, indeterminate spores, and fungi are scarce.

### 4.4.2 Zone II (300 to 220 cm, sample 36-44)

This zone is characterized by drastic reduction of marine palynomorphs with respect to the previous zone (e.g., microforaminiferal linings, acritarchs and *Cymatiosphaera*) and the disappearance of dinoflagellate cysts. Freshwater algae assemblage remains nearly unchanged, however *Botryococcus* increases its percentages through this zone (1-22 %). The terrestrial palynomorphs are dominated by herbs (41-65 %), followed by trees and shrubs (31-54 %). Ferns have a relative increase (13–25 %) and indeterminate pollen grains have a relative decrease (2–5 %) with respect to the zone I. Epiphytes,

lianas and climbers, bryophytes, indeterminate spores, fungi, and algae maintain relative lower abundances.

### 4.4.3 Zone III (210 to 80 cm, sample 45-58)

This zone is marked by the reduction of freshwater algae (<8 %) and the disappearance of *Botryococcus*, *Pseudoschizaea rubina* and *Zygnema*. The marine palynomorphs were not recorded. Fungi increase their percentages (2–24 %) as well as herb pollen grains (55–92 %). These latter are represented mainly by Poaceae, Asteraceae subf. Asteroideae and Cyperaceae.

Additionally, bryophyte spores are mainly represented by species of *Phaeoceros*, which occur significantly for the first time in the core reaching up to 13 %. Trees and shrubs decrease their relative abundances (8–42 %), as well as the indeterminate pollen grains (<2 %). Similarly to the previous zone, epiphytes, lianas and climbers, bryophytes and indeterminate spores have low relative abundances.

### 4.4.4 Zone IV (70 to 0 cm, sample 59-66)

This zone comprises the upper portion of the core. It is characterized by an increase of pollen sum of arboreal taxa (34–86 %) and epiphytes, lianas and climbers (0.3-10 %). Herb pollen taxa decrease their relative abundances (12–73 %), as well as bryophyte taxa (<17 %). The arboreal assemblage is mainly represented by Myrtaceae, Arecaceae and *Ilex;* epiphytes, lianas





and climbers are represented by Cucurbitaceae. Pteridophytes show an increase of their percentages and are represented mainly

by *Blechnum* and Polypodiaceae taxa. Indeterminate pollen grain and indeterminate spores maintain low relative abundances.

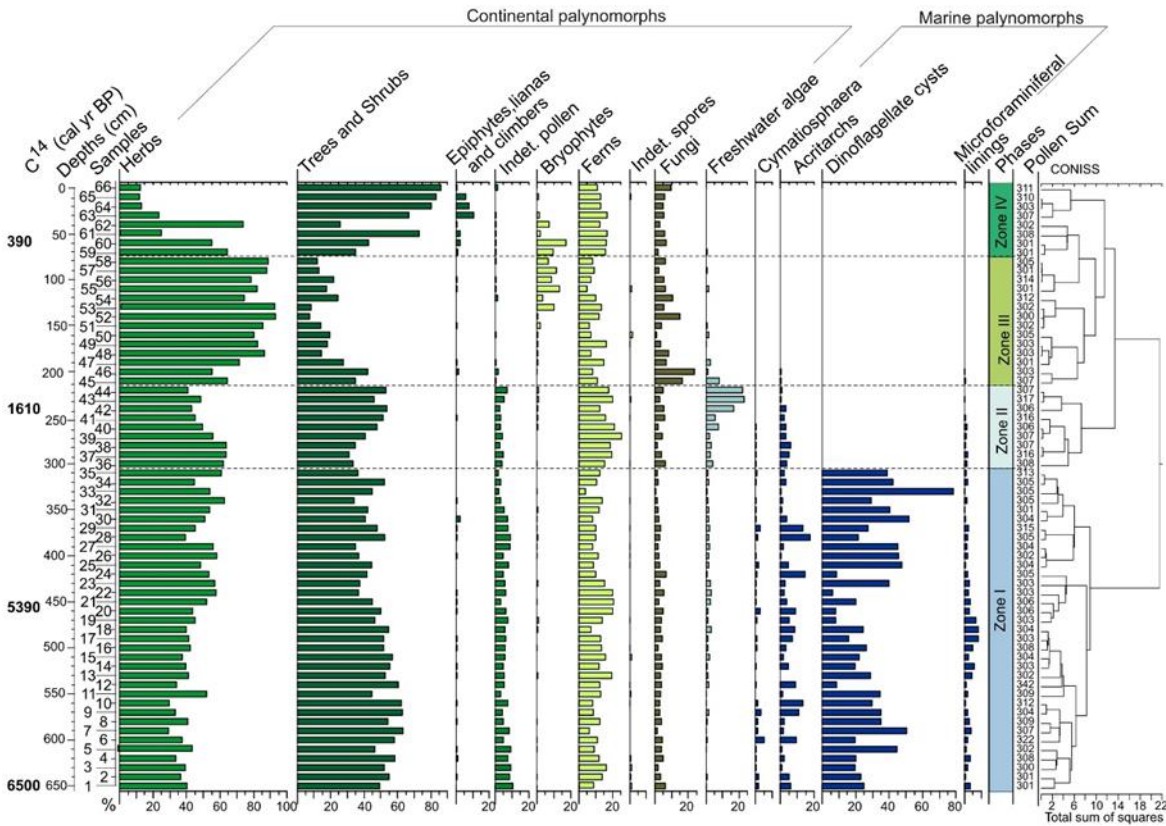


**Fig. 5.** Percentage diagram of the palynomorphs grouped according to their ecological affinities (habit or habitat) from the

PCSC-4 core, south of Santa Catarina Island, southern Brazil, as well as the identified zones and cluster analyses.



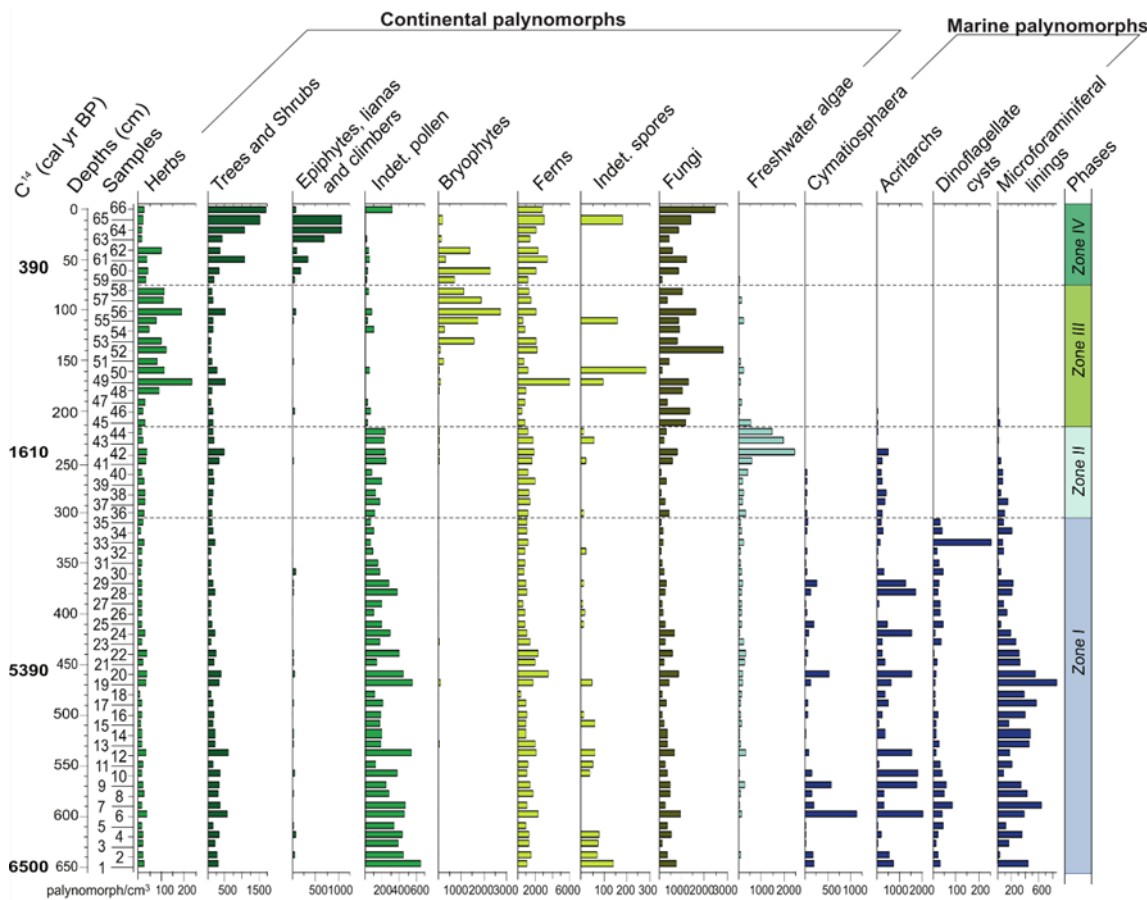

**Fig. 6.** Concentration diagram of the palynomorphs grouped according to their ecological affinities (habit or habitat) from the
PCSC-4 core, south of Santa Catarina Island, southern Brazil, as well as the identified zones.



**Fig. 7.** Relative abundance diagram of the palynomorphs taxa according to their ecological affinities (habit or habitat) from the PCSC-4 core, south of Santa Catarina Island, southern Brazil, as well as the identified zones.





**Fig. 8.** Relative abundance diagram of the dinoflagellate cysts taxa (a) and concentration diagram of the dinoflagellate cysts taxa (b) from the PCSC-4 core, south of Santa Catarina Island, southern Brazil.



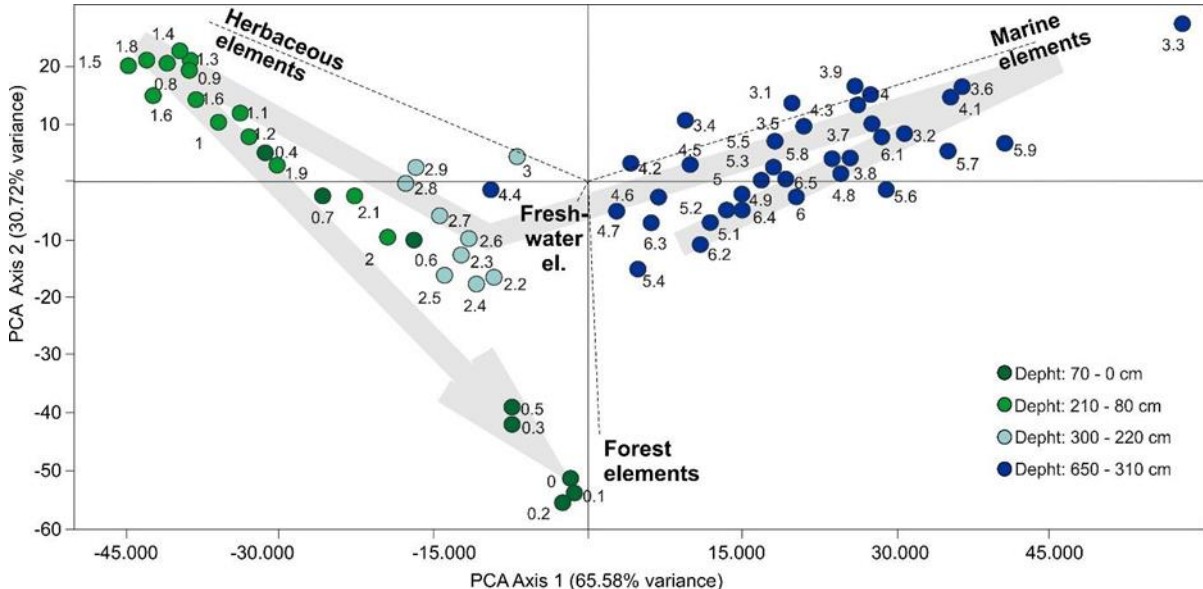

**Fig. 9.** Biplot of the two main PCA axes with indication of palynomorph groups ordination. Numbers next to the circles indicate the depths of the samples (m).

## 5 Discussion

### 5.1 Zone I: Lagoonal stage (6500 cal yr BP–2820 cal yr BP)

The predominance of fine sand, silt and clay sediments, as well as the presence of preserved calcareous shells in living position, indicate deposition in a predominantly calm water body. High percentages of marine palynomorphs, including dinoflagellate cysts (8–78 %), microforaminiferal linings (<8 %), acritarchs (<17 %) and *Cymatiosphaera* (<5 %) are evidence of sea waters reaching the sampling site.

Microforaminiferal linings are abundant in estuarine marshes with variable salinity water influence (Batten, 1996). Species of the acritarch genus *Micrhystridium* are characteristic of shallow coastal water associations (Montenari and Leppig, 2003; Félix and Souza, 2012). The prasinophyte *Cymatiosphaera* on the other hand is known to be typically associated with marine water (Mudie et al., 2020).

The dinoflagellate cysts include photosynthetic (*Operculodinium centrocarpum*, *O. israelianum*, *Spiniferites* spp., *S. mirabilis*, *P. dalei*) and heterotrophic taxa (*Brigantedinium* spp., *Leipokatium invisitatum*, *Polykrikos kofoidii/schwarzii*, *Protoperidinium* spp., *Selenopemphix nephroides*; Fig. 8). All the above-mentioned dinoflagellate cysts are commonly registered in the South Atlantic Ocean, as well as in coastal sites (Zonneveld et al., 2013). The presence of heterotrophic taxa in the association suggests high nutrient inputs in the waters. The photosynthetic association is dominated by *Operculodinium centrocarpum* and *Spiniferites* spp. On a global scale, the species *Operculodinium centrocarpum* has a cosmopolitan distribution (e.g., Zonneveld et al., 2013). However, in the western South Atlantic, this species is typically present in the




relatively warm waters of the BC (Gu et al., 2019). Throughout this zone, specimens of *Operculodinium israelianum* and *Spiniferites mirabilis* are registered, which are typical for warm, temperate waters (Zonneveld et al., 2018). The dinoflagellate cyst association therefore indicates that marine waters entering in the study site had their origin in the relatively warm saline
BC waters.

Even though nowadays waters of the BCC/ MC can seasonally reach the coast at the same latitude of the core position, we did not observe any evidence that this has been the case in zone I (Fig. 2). Species that are characteristically abundant in MC and BCC waters, such as *Selenopemphix antarctica* and/or *Impagidinium variaseptum* (Zonneveld et al., 2013; Gu et al., 2019), were not observed in the studied material. The presence of a warm, temperate dinoflagellate cysts association, characteristic
for high nutrient concentrations, implies that the lagoon waters were relatively warm and nutrient rich during this zone.

The presence of freshwater algae indicates freshwater influence despite the significant marine contribution. In addition, *Botryococcus* is a euryhaline freshwater algae that may have its photosynthetic activity inhibited, directly or indirectly, by the water salinity (Tyson, 1995). Therefore, the low concentrations of this freshwater algae during this zone might be caused by the presence of marine influence. The observed palynomorph association suggests that calm conditions and mixing of marine
and freshwater prevailed typical for a lagoon environment.

Pollen grains (herbs, trees, and shrubs ) were transported by streams and/or wind and deposited in the lagoon in low concentrations (Fig. 6). The herbs taxa are mostly represented by Poaceae, *Amaranthus*/Chenopodioideae, Apiaceae, Asteraceae subf. Asteroideae and Cyperaceae (Fig. 7). These herbs are better adapted to sandy soils and include many halophytes (Lorscheitter, 2003). This suggests that saline soil conditions prevailed in the margins of the lagoon.
In this zone, the pollen of the trees and shrubs taxa are mainly composed by *Alchornea*, Arecaceae, Myrtaceae and *Celtis*. These taxa are indicators of the Atlantic Rainforest (Lorscheitter, 2003) and the occurrence of these taxa suggests the occurrence of the Atlantic Rainforest in the surrounding area, most likely in the hills around the lagoon (Fig. 10; Zone I). Indeterminate pollen grains occur at higher concentrations at this and the subsequent zone II in comparison to the two upper zones. This can be explained by the input of pollen grains transported by streams and wind into the lagoon partly damaging
some of the grains. The pollen grains that originated from the Andes (*Nothofagus* and *Alnus*), are expected to be long-distance transported by air dispersion to the deposition site. These palynomorphs have been previously found in many Quaternary sedimentary profiles from southern Brazil coastal plain (e.g., Cancelli et al., 2012; Diniz and Medeanic, 2012; Masetto and Lorscheitter, 2016; Kuhn et al., 2017; Silva et al., 2021).

The combination of high $\delta^{13}C$ values (-19 to -12) and low C/N (14–16) ratios indicate mixtures of terrestrial and marine
palynomorphs and show little variation throughout this zone (Figs. 3, 4). Low C/N ratios are attributed to the presence of nitrogen-enriched freshwater algae organic matter (Meyers, 1994; 1997; Wilson et al., 2005). Among terrestrial plants, higher $\delta^{13}C$ values indicate predominance of $C_4$ plants over $C_3$ plants. This can be explained by the deposition of herbs remnants from the borders of the lagoonal body. TOC values do not exceed 10 %. Such low TOC values are typical of lagoonal and estuarine environment (Tyson, 1995; Lorente et al., 2014).





## 5.2 Zone II: Regressive stage and sea disconnection ( 2820 cal yr BP–1480 cal yr BP)

In this zone, predominance of fine sediments and the occurrence of freshwater algae and marine palynomorphs still indicates the presence of a calm brackish water body (Fig. 10; Zone II). However, the significant reduction of the marine indicators (e.g., microforaminiferal linings, acritarchs and *Cymatiosphaera*) and the disappearance of dinoflagellate cysts point out to a progressive reduction of sea water input into the water body. Towards the end of this zone (ca. 1480 cal yr BP) the lagoon was probably disconnected from the sea (Fig. 10; Zone II). This is supported by the observation of the increasing in abundance of *Botryococcus* towards the top of the zone, suggesting that freshwater inputs into the lagoon progressively decreased its salinity transferring the lagoon into a freshwater lake. Previous studies suggest that an increase of *Botryococcus* concentration might be related to a decrease of the water level in a lake (Tyson, 1995). It is therefore likely that during the process of the closing of the lagoon and freshening waters, water table dropped consistently.

No significant changes were observed in the spore-pollen assemblage. However, the decrease of *Amaranthus*/Chenopodiaceae throughout this zone suggests a progressive desalination of the soil of the adjacent areas of the lagoonal body. The C/N ratios remain at low values of ca. 15 % throughout this zone, indicating that freshwater algae are still major contributors to the total organic matter. The binary plot of $\delta^{13}C \times$ C/N shows a trend towards lower $\delta^{13}C$ values likely related to the increase of freshwater phytoplankton input.

## 5.3 Zone III: Early development of the Restinga Forest (coastal plain Atlantic Rainforest) (1480 cal yr BP–520 cal yr BP)

The transition from Zone II to III is marked by a drastic decline of freshwater algae concentrations and the disappearance of *Botryococcus*, *Pseudoschizaea rubina* and *Zygnema*. This reduction suggests a reduced fresh-water input into the area. Moreover, marine palynomorphs are no longer recorded in this zone because of the disconnection with the sea that occurred in the previous zone.

The development of soils rich in organic matter at the site is evidenced by the increase of fungi. The abundance of fungal fragments is indicative of aerobic biodegradation of plant remains (Sebag et al., 2006). Additionally, the increase of the organic matter in the sediment and the high values of TOC (average of ca. 44 %) indicates the development of soils rich in organic matter during this zone.

High percentages of Poaceae and Cyperaceae taxa and $\delta^{13}C$ enrichment from the beginning of this zone up to 140 cm of depth can be observed in this zone (Figs. 3, 7). This suggests that the herbs that previously occupied the margins of the lagoon advanced and colonized the palaeo-lagoon area and the environment of ongoing humid soil condition. The concentration values for trees and shrubs remain constant throughout this zone suggesting that input of arboreal pollen that were transported from adjacent areas covered by Atlantic Rainforest did not change (Fig. 10; Zone III). Consequently, the observed decrease of relative abundances of trees and shrubs associated with increasing herbs supports our previous suggestion of a significant increase in the development of herbs in the palaeo-lagoon. The strong increase in the C/N ratios (Fig. 3) in this zone indicates an input of carbon-enriched material and suggests the increase of dense vegetation in the adjacent areas. The C/N ratios greater

 

than 20 are originated from vascular land plants (Meyers, 1994) and δ¹³C values around -25‰ indicate an influence of C₃ plants (Meyers, 1997). The binary plot of δ¹³C × C/N also shows that the organic matter was influenced by C₃ plants (Fig. 4).
In the upper part of the section related to this zone, Poaceae and Cyperaceae (herbs) decreased their relative abundances whereas Myrtaceae and Arecaceae (trees and shrubs) increase, showing a succession from herbal to arboreal taxa of the Restinga Forest.

**5.4 Zone IV: Restinga Forest (coastal plain Atlantic Rainforest) (520 cal yr BP–present)**

This zone is characterized by a decrease in relative abundances of herbs in favor of trees and shrubs. This suggests that forests
developed in the palaeo-lagoon. The high relative abundances of arboreal taxa typical of the Atlantic Rainforest (Arecaceae, *Ilex* and Myrtaceae) marks the consolidation of this arboreal forest in the area during this zone. In addition, the increase of epiphytes, lianas and climbers (mainly Cucurbitaceae taxa) suggest the advance of arboreal components in the vegetational-succession of the forest as well (IBGE, 2012; Sevegnani and Schroeder, 2013).
Pteridophytes that are typically associated with the arboreal Restinga Forest (*Blechnum* and Polypodiaceae; Falkenberg, 1999)
increase their concentration and supports the consolidation of the arboreal Restinga Forest in the region. The isotope data indicates a depletion in the δ¹³C values during this zone, reaching the lowest value in the uppermost sample (-30). These δ¹³C values reveal the dominance of C₃ plants, also indicated in the pollen record. In addition, the high TOC values and abundant organic matter contents in the sediment (~40 and ~90 %, respectively) indicate the occurrence of a dense forest similar to the one that cover the area nowadays.





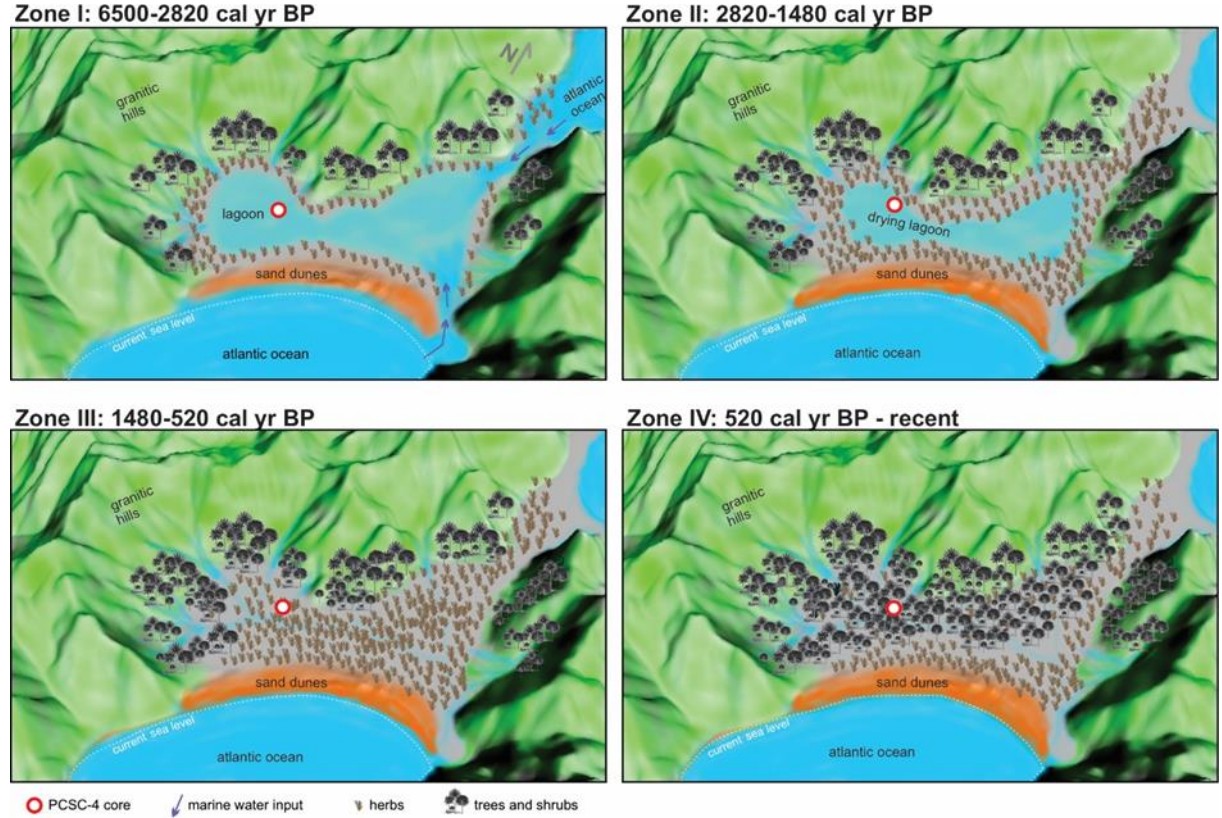


**Fig. 10.** Schematic model of the environmental evolution of the southernmost portion of the south of Santa Catarina Island (Pântano do Sul), southern Brazil.

**5.5 Environmental evolution of southern Brazil coastal plain and regional implications**

The observed transition from a lagoonal environment at ca. 6500 cal yr BP to the actual Restinga Forest, in the Pântano do Sul

area, is interpreted to be directly related to sea level changes throughout the Holocene. In general, there is an observed sequence of decreasing marine water contribution that indicates that the relative sea level was higher than the current sea level in the first zone successively decreasing during the second zone. Because of the disconnection of the sea and the lagoonal body evidenced by the absence of marine palynomorphs in Zone III and Zone IV, we interpret that the sea level reached the current level at some point during one of these zones, most likely at Zone III. Previous studies on the Brazilian coastal plain suggested

the existence of high-frequency oscillations in the relative sea level with two regressive zones during the late Holocene (Suguio et al., 1985; Martin et al., 2003). The authors suggested that the sea level was slightly below its present elevation at ca. 4200-3700 and 2700-2100 yr BP. However, recent studies suggest a regular decline in the relative sea level without significant oscillations during the late Holocene (Angulo et al., 1999; 2006; 2022; Ybert et al., 2003). In particular, Angulo et al. (2006; 2022) suggested that the highstand in the Holocene occurred between 5000 and 5800 yr BP without a distinct peak. The phase

succession observed in our study can be better explained by a regular decline without significant oscillations as proposed by



Angulo et al (2006; 2022; Fig. 11). Moreover, other palynological studies in the southern Brazil coastal plain in the Rio Grande do Sul (e.g., Cordeiro and Lorscheitter, 1994; Lorscheitter and Dillenburg, 1998; Meyer et al., 2005; Masetto and Lorscheitter, 2019) and Santa Catarina states (Behling and Negrelle, 2001; Amaral et al., 2012; Cancelli, 2012; Kuhn et al., 2017; Val-Peón et al., 2019; Cohen et al., 2020; Silva et al., 2021), also identified the marine influence ca. 6000-5000 yr BP at their sites and

showed a similar sea-level dynamic. In addition, palynological studies of palaeoenvironmental reconstitutions performed in the coastal areas of Uruguay and Argentina also indicated a highstand sea level between ca. 6000-5000 yr BP followed by a regressive event (e.g., Borel and Gómez, 2006; García-Rodríguez et al., 2010; Mourelle et al., 2015; Vilanova and Prieto, 2012).

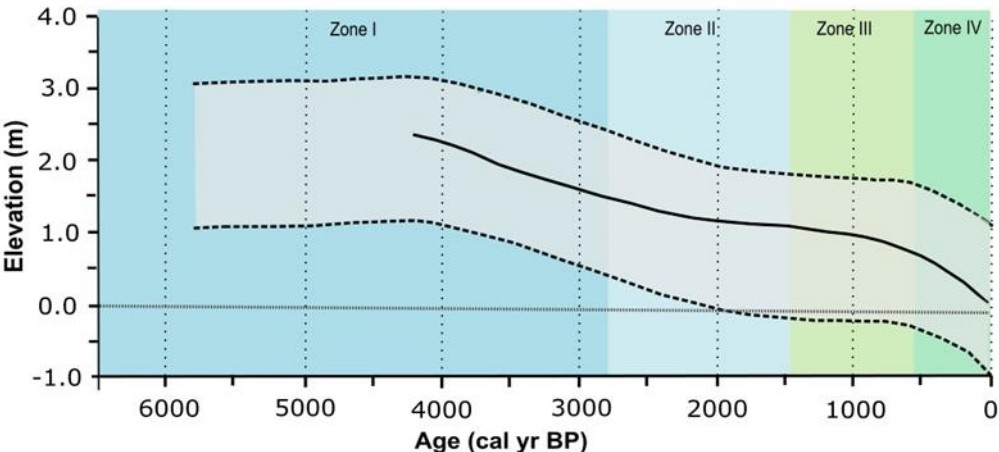

**Fig. 11.** Paleo-sea level reconstructions for the southern Brazilian coast and paleoenvironmental zones. Gray area: relative paleo-sea level envelope (Angulo et al., 2006) to the south of 28°S latitude; Solid line: sea-level model curve for Ponta do Papagaio (Angulo et al., 2022) (modified from Angulo et al., 2022).

In general, the environmental zone succession identified in this study is similar to those presented in previous
palaeoenvironmental studies developed on the southern Brazil coastal plain, in particular in the Santa Catarina sector (e.g., Amaral et al., 2012; Cancelli, 2012; Kuhn et al., 2017; Val-Peón et al., 2019; Silva et al., 2021). Most of these studies indicate a succession of three zones from a lagoonal/estuarine environment (I) to a transitional/swampy regime (II) and an arboreal forest environment (III). However, in this contribution we were able to define the transition from an herbaceous Restinga (Zone III) to the arboreal Restinga Forest (Zone IV). Some of the above-mentioned studies indicate a maximum RSL ranging
from ca. 5200 to 4500 yr BP (Fig. 12).

The distinct ages for the end of marine influence and the development of the Atlantic Rainforest described in the previous studies is probably related to the different distance and/or altitude of the depositional sites in relation to the current coastline and sea level. Localities nearer to the sea (e.g., Kuhn et al., 2017 and this study) were more affected by the sea level rise and





show late development of the Atlantic Rainforest in comparison to those located further from the shoreline (e.g., Cancelli, 2012). This supports that the Restinga Forest development was mainly controlled by edaphic factors and less sensitive to climate factors during the Holocene (Scheel-Ybert, 2000; Amaral et al., 2012; Melo Jr. and Boeger, 2015).





**Fig. 12.** Palynological studies and their location. (a) Summarized environmental changes of the palynological studies in similar settings of the Santa Catarina coastal plain. (b) Sites from the Rio Grande do Sul State, Uruguay (UY) and Argentina (AR).



(c) Sites from the Santa Catarina State (in Fig. 12a). Max. RSL = Maximum relative sea level. Inf. = Influence. Cross bars at lower limits indicate that the core was older than 6500 cal yr BP.

## 6 Conclusion

Palynological, stable isotopic and sedimentological analyses at the southernmost Santa Catarina Island allowed us to recognize four environmental zones from 6500 cal yr BP to present namely: Zone I (lagoon with sea influence), Zone II (lagoon without
sea influence), Zone III (early development of the Restinga Forest) and Zone IV (Restinga Forest). The dinoflagellate cyst association suggests that marine waters entering the region had their origin in the relatively warm saline BC waters. Even though nowadays waters of the Brazil Coastal Current/MC can seasonally reach the coast at the same latitude of the core position, we did not observe any evidence that this has been the case in zone I. Pollen records indicate that the Atlantic Rainforest was already present in the Santa Catarina coastal plain before 6500 cal yr BP, however likely restricted to areas
which were not affected by the maximum transgressive sea level in the Holocene. Furthermore, we observed that the development of the Restinga Forest in the area occurred subsequently to the drying up of a lagoon. This study enhances our knowledge of the evolution of southern Brazil coastal plain through information of geomorphological and vegetational changes during the Holocene. Furthermore, this study represents an example of the strong sensitivity of southern Brazilian ecosystem change caused by relative sea level variations. As such, it contributes to the debate about potential effects of current climate
change induced by global sea level changes.

### Data availability

All raw data can be provided by the corresponding authors upon request.

### Author contributions

LAK: Conceptualization, Methodology, Formal analysis, Investigation, Writing- Original draft, Writing - Review & Editing,
Funding acquisition; KAFZ: Conceptualization, Methodology, Formal analysis, Resources, Writing - Review & Editing; Supervision; Funding acquisition; PAS: Conceptualization, Methodology, Resources, Writing - Review & Editing, Supervision, Project administration; RRC: Conceptualization, Formal analysis, Investigation, Writing - Review & Editing.

### Compelling interests

The authors declare that they have no conflict of interest.



## Acknowledments

The authors thank Pedro H. Simas for fieldwork assistance and guidance at the study area, to Guilherme S. Hoerlle for fieldwork and writing assistance, and Beatriz Fontana for the English revision. This paper is part of the LAK PhD Thesis developed at the Programa de Pós-graduação em Geociências, Universidade Federal do Rio Grande do Sul.

Funding: this work was supported by the CNPq-National Council for Scientific and Technological Development of Brazil (grant number 141324/2017) and CAPES-Brazilian Coordination of Higher Education Staff Improvement (grant number 88887.467306/2019-00).

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
