# Peer review of "Late Quaternary palaeoenvironmental evolution and sea level oscillation of the Santa Catarina Island (southern Brazil)"

_Biogeosciences, 2023_

## Author Response (AR2)

Dear Dr Petr Kuneš,

On behalf of the co-authors, I thank you for accepting our manuscript "Late Quaternary palaeoenvironmental evolution and sea level oscillation of the Santa Catarina Island (southern Brazil)" for publication in Biogeosciences.

I hereby submit the revised version of our manuscript according to the reviewers' comments as well as your suggestions. We indicated in detail our actions below:

**Comments from the editors and reviewers:**

**Associate editor decision: Publish subject to minor revisions (review by editor Petr Kuneš**

Dear authors,

Two reviewers and myself have evaluated the manuscript and find it worth publishing in Biogeosciences. However, before I can accept the manuscript for publication, it requires some minor to moderate changes. Please follow the reviewers' comments and submit your revision as soon as possible.

**Ans.:** We are happy to learn that the reviewers and the editor recognized the relevance of our contribution. We thank the editor and reviewers for the careful and relevant comments.

I have a few comments myself. First, I have some doubts concerning your age-depth model and application of calibrated ages. In the methods you mention that you first calibrated the radiocarbon ages and then used linear function to interpolate sample ages. You report the calibrated ages as an interval, but in Fig 3 I only see points. What are these points? Instead, I suggest you perform a rigorous age-depth modelling following any of the state-of-the-art techniques: Oxcal, Clam, Bacon, Bchron…

**Ans.:** We performed the age-depth modelling as recommended by the editor. The radiocarbon dates were calibrated using the Southern Hemisphere calibration curves (SHCal20; Hogg et al., 2020), and the age-depth model was

constructed using Clam 2.4.0 package (Blaauw, 2010) with linear interpolation. Therefore, we added a new figure (Figure 3) showing the age-depth model.

Secondly, our data policy explicitly says "Copernicus Publications requests depositing data that correspond to journal articles in reliable (public) data repositories, assigning digital object identifiers, and properly citing data sets as individual contributions."

I therefore ask you to upload your data to any of the public repositories, and provide DOI's upon publication of your manuscript. I suggest you contact representatives of the Latin American Pollen Database (LAPD), which is now hosted in the Neotoma database, and request your data upload.

Let me know if you have any troubles with uploading your data.

**Ans.:** We deposited our data in the Neotoma database. The data can be accessed through this permanent link: https://data.neotomadb.org/55958
We include the permanent link to our data in Neotoma (https://data.neotomadb.org/55958) in the final version of our manuscript. When available, the DOI can be accessed on the page of the permanent link itself.

**Hermann Behling**

**(Behling comments**: https://doi.org/10.5194/bg-2023-11-RC1; **Kuhn et al comments**: https://doi.org/10.5194/bg-2023-11-AC1)

The manuscript by Lidia A. Kuhn and co-authors provides important new results on vegetation and environmental changes in the coastal region of southern Brazil since the last about 6500 years. This multi-proxy study is of high international interest from the coastal region which has been poorly studied. The manuscript is well prepared. Following smaller points should be considered.

**Line 13:** Take "relative" out

**Ans.:** Corrected as suggested.

**Line 17:** Change "in" to for

**Ans.:** Changed as suggested.

**Line 46: "**paleo…" should be palaeo…. , be consistent with UK English, see also other parts of the manuscript

**Ans.:** Corrected as suggested.

**Line 55:** "only"? There is also a marine core studied off Santa Catarina State: Gu, F., Zonneveld, K.A.F., Chiessi, C.M., Arz, H..W., Pätzold, J., Behling, H., 2017. Long-term vegetation, climate and ocean dynamics inferred from a 73,500 years old marine sediment core (GeoB2107-3) off southern Brazil. Quaternary Science Reviews, 172C, 55-71. DOI information: 10.1016/j.quascirev.2017.06.028

There are also more studies from the coastal region in Santa Catarina. See e.g. Behling, H., 1995. Investigations into the Late Pleistocene and Holocene history of vegetation and climate in Santa Catarina (S Brazil). Vegetation History and Archaeobotany, 4, 127-152.

**Ans.:** We added the respective references and modified the manuscript to: "*However, the Santa Catarina coastal plain sector is geomorphologically distinct and similar studies are scarce, located only in the continental portion (Behling, 1995; Behling and Negrelle, 2001; Amaral et al., 2012; Cancelli, 2012; Kuhn et al., 2017; França et al., 2019; Val-Péon et al., 2019; Cohen et al., 2020; Silva et al., 2021) and at the continental slope at the western South Atlantic (ca. 200 km north of the Santa Catarina Island; Gu et al., 2017*)."

**Line 163:** Does it make sense to have a sum of all palynomorphs? Such a curve is not meaningful.

**Ans.:** We determined the relative abundances of the individual palynomorphs groups from terrestrial and marine origin by dividing of with the total palynomorph sum to obtain insight into the relative change of marine and terrestrial derived elements over time. To obtain insight into the contemporaneous vegetation changes on land and the marine environmental changes we determined the relative abundances of the individual pollen/spore species and dinoflagellate cyst species by dividing by the total pollen/spores sum and the dinoflagellate cyst sum respectively.

We realized that the text was quite confusing written in this respect and therefore adapted the text such that it is clearer.

We modified the text to:

"*To obtain information about what part of the palynomorph association consist of terrestrial derived elements (pollen/spores, freshwater algae), fungi and marine elements (acritarchs, dinoflagellate cysts and microforaminiferal linings), we used the total sum of all palynomorphs to determine the relative abundances of these groups. Relative abundances of pollen/spores are determined by dividing the pollen counts by the pollen sum referring the total amount of pollen grains. The relative abundances of dinoflagellates were calculated by dividing their counts by the total dinoflagellate cyst sum.*"

**Line 167:** "total sum" of all palynomorphs? Does it make sense to use all, as they reflect very different environments?

**Ans.:** See later point

**Line 307:** Should be Chenopodiaceae.

**Ans.:** Corrected as suggested.

It would be good to indicate the number of counted dinoflagellate cysts for each sample.

**Ans.:** We included information about the total amount of dinoflagellate cysts counted in Figure 9 and we will provide all palynological data as a supplementary data.

**Aline Freitas**

**(Freitas comments:** https://doi.org/10.5194/bg-2023-11-RC2 and **Kuhn et al comments**: https://doi.org/10.5194/bg-2023-11-AC2**)**

**Page 17, line 302** you cite *Botryococcus* as freshwater algae that tolerate little salinity, but by the way of a complementary information and discussion, Batten & Grenfell (1996)* cite its occurrence in calm or stagnant waters, such as lakes, swamps, marshes, marshes, although it can withstand relatively higher salinity from other environments such as mangroves and estuaries. It can be found in shallow or deep bodies of water.

**Ans.:** We included these comments in the manuscript and modified the manuscript to: "*The presence of freshwater algae indicates freshwater influence*

*despite the significant marine contribution. In addition, Botryococcus is a euryhaline freshwater algae that may have its photosynthetic activity inhibited, directly or indirectly, by the water salinity (Tyson, 1995). Also, Batten and Grenfell (1996) cite Botryococcus occurrence in calm or stagnant waters, such as lakes, swamps, marshes, although it can withstand relatively higher salinity from other environments such as mangroves and estuaries. Therefore, the low concentrations of this freshwater algae during this zone might be caused by the presence of marine influence. The observed palynomorph association suggests that calm conditions and mixing of marine and freshwater prevailed typical for a lagoon environment*".

**Page 17, line 310** the presence of *Celtis* and *Trema* (Zone I, Fig. 7), pioneer taxa in this phase, helps the inference of an unstable environment, subject to ecological succession. And together with the shrub-tree taxa (*Alchornea*, Myrtaceae and Arecaceae) and halophytes (Amaranthaceae-Chenopodiaceae and others) it indicates seasonally flooded open restinga, with sandy soils subject to salinity (see *Allagoptera* that occurs in the Holocene coastal sediments of Armação dos Búzios, in Southeastern Brazil**). The presence of *Botryococcus* confirms this hypothesis.

**Ans.:** We included these comments in the manuscript and changed the manuscript to: "*In this zone, the pollen of the trees and shrubs taxa are mainly composed by Alchornea, Arecaceae, Myrtaceae and Celtis. The presence of Celtis and Trema, pioneer taxa, suggests an unstable environment, subject to ecological succession. These pioneer taxa together with the trees and shrubs taxa (Alchornea, Myrtaceae and Arecaceae) and the halophytes herbs taxa (e.g., Amaranthus/Chenopodiaceae) indicate seasonally flooded open restinga in the surrounding area, with sandy soil subject to salinity (Freitas and Carvalho, 2012)*".

**Page 17, line 313** the presence of indeterminate pollen in this phase may be related to the transport of organic particles and the granulometry of sediments (fine sands and silts), in the very redeposition of lacustrine-lagoonal sediments configuring a high-energy and oxygenated environment, under greater influence of currents marine (mentioned in the Fig. 2) than continental (terrestrial biomass) contribution. What the assemblage of dinoflagellates and other marine organic microfossils attest this inferences.

**Ans.:** We included the reviewer comment as a possible interpretation and modified the manuscript to "*Indeterminate pollen grains occur at higher concentrations at this and the subsequent zone II in comparison to the two upper zones. This can be explained by the input of pollen grains transported by streams and wind into the lagoon partly damaging some of the grains. Alternatively, the presence of indeterminate pollen in this phase may be related to the transport of organic particles and the granulometry of sediments (fine sands and silts), in the very redeposition of lacustrine-lagoonal sediments configuring a high-energy and oxygenated environment, under greater influence of marine currents than continental (terrestrial biomass) contribution. However, the predominance of fine sand, silt and clay sediment, as well as the presence of preserved calcareous shells in living position, could indicate deposition in a predominantly calm water body. Therefore, we interpret that the water body was likely calm with sporadic higher energy events (e.g., storms)*".

**Page 18, lines 336** for complementation purposes, the expressive presence of *Botryococcus* may indicate marine water inputs in the lake/lagoonal body, even with a tendency to clog, since these microalgae are indicators of lake-lagoon environments, with a certain tolerance to salinity (Batten & Grenfell, 1996).

**Ans.:** In this phase the increase in *Botryococcus* algae coincides with the decrease in marine palynomorphs and the level that there are the highest percentages of *Botryococcus* almost there aren´t marine palynomorphs (see figure 6). Dinoflagellate cysts are no longer present in this phase. In addition, there is an increase in *Botryococcus* in the middle of this zone and a decrease in the end of zone I and beginning of zone II. So, it doesn't look like salinity pulses in the lagoon body.

**Page 18, lines 344 and 350** dispite ecological data of *Zygnema* spores, they are cosmopolitan in oligotrophic lacustrine, shallow and clear freshwater environments. Also present in fertile soils, and lakes and peat bog margins. A few members of Zygnemataceae can withstand more adverse environmental conditions, such as seasonal droughts and higher temperatures during winter (van Geel & Grenfell 1996\*\*\*). Cyperaceae it is an emergent aquatic plant common in shallow or deep lagoons or salt marshes.

**Ans.:** We included the reviewer comment regarding Cyperaceae and modified the manuscript to: "*High percentages of Poaceae and Cyperaceae taxa and δ¹³C enrichment from the beginning of this zone up to 140 cm of depth can be observed in this zone (Figs. 3, 7). In particular, Cyperaceae is an emergent aquatic plant common in restinga vegetation in lagoon environments, marshes and swampy lowlands (Falkenberg, 1999). This suggests that the herbs that previously occupied the margins of the lagoon advanced and colonized the palaeo-lagoon area and the environment of ongoing humid soil condition*." However, we did not include the comment about *Zygnema* because this taxa was not recorded in this zone and in the previous zone it was recorded in a low percentages.

***Page 20, lines 384*** I would add that the *Botryococcus* microalgae in Phase/Zone III indicate the salinity pulses in the lacustrine-lagoon body.

**Ans.:** We understand the reviewer's comments and we acknowledge that *Botryococcus* withstands saline waters. However, given the trend of disappearance of marine palynomorphs followed by a terrestrial environment we consider that it is more likely that the increase in *Botryococcus* algae in zone II is related to a decrease of the water level in a lake.

**Aline Freitas**

**(Freitas comments:** https://doi.org/10.5194/bg-2023-11-RC3 and **Kuhn et al comments**: https://doi.org/10.5194/bg-2023-11-AC3)

I am grateful for the considerations on the part of the authors, and I consider the paper strongly suitable for publication.

**Ans**.: We thank Dr. Aline Freitas for recognizing the relevance of our contribution and we thank her for her careful review.

**References:**

Batten DJ, Grenfell HR. 1996. Botryococcus. In: Jansonius J, McGregor DC. (eds.) Palynology: principles and applications. Dallas, American Association of Stratigraphic Palynologists Foundation. Vol. 1. p. 205-214.

Behling, H., 1995. Investigations into the Late Pleistocene and Holocene history of vegetation and climate in Santa Catarina (S Brazil). Vegetation History and Archaeobotany, 4, 127-152

Blaauw, M.: Methods and code for 'classical'age-modelling of radiocarbon sequences. Quat geochronology, 5, 512-518, doi: https://doi.org/10.1016/j.quageo.2010.01.002, 2010.

Freitas & Carvalho, 2012 Freitas AG, Carvalho MA. 2012. Análise morfológica e inferências ecológicas de grãos de pólen e esporos (últimos ~8.000 anos) da Lagoa da Ferradura, Armação dos Búzios, RJ, Brasil. Revista Brasileira de Paleontologia 15: 300-318.).

Gu, F., Zonneveld, K.A.F., Chiessi, C.M., Arz, H.W., Pätzold, J., Behling, H., 2017. Long-term vegetation, climate and ocean dynamics inferred from a 73,500 years old marine sediment core (GeoB2107-3) off southern Brazil. Quaternary Science Reviews, 172C, 55-71. DOI information: 10.1016/j.quascirev.2017.06.028

Hogg, A.G., Heaton, T.J., Hua, Q., Palmer, J.G., Turney, C.S., Southon, J., Bayliss, A., Blackwell, P.G., Boswijk, G., Ramsey, C.B. Pearson, C., Petchey, F., Reimer, P., Reimer, R. and Wacker, L.: SHCal20 southern hemisphere calibration, 0-55,000 years cal BP, Radiocarbon, 62, 759-778, doi: 10.1017/RDC.2020.59, 2020.

Van Geel B, Grenfell HR. 1996. Spores of Zygnemataceae. In: Jansonius J, McGregor DC. (eds.) Palynology: principles and applications. Dallas, American Association of Stratigraphic Palynologists Foundation. Vol 1. p. 173-179.